# Random forests, sound symbolism and Pokémon evolution

**Alexander James Kilpatrick**[1]*, **Aleksandra Ćwiek**[2], **Shigeto Kawahara**[3]

**1** International Communication, Nagoya University of Commerce and Business, Nagoya, Aichi, Japan,
**2** Department Leibniz-Zentrum Allgemeine Sprachwissenschaft, Berlin, Germany, **3** Institute of Cultural and
Linguistic Studies, Keio University, Tokyo, Japan

☉ These authors contributed equally to this work.
* alexander_Kilpatrick@nucba.ac.jp

Random forests, sound symbolism and Pokémon
evolution. PLoS ONE 18(1): e0279350. https://doi.
org/10.1371/journal.pone.0279350

Communications, JAPAN

**Data Availability Statement:** All data, scripts and
an explanation of their implementation are available
at the following OSF repository. https://osf.io/
pe24w/?view_only=
02e9327a7bd54b9280b57434a90ed83a

## Abstract

This study constructs machine learning algorithms that are trained to classify samples using
sound symbolism, and then it reports on an experiment designed to measure their under-
standing against human participants. Random forests are trained using the names of Poké-
mon, which are fictional video game characters, and their evolutionary status. Pokémon
undergo evolution when certain in-game conditions are met. Evolution changes the appear-
ance, abilities, and names of Pokémon. In the first experiment, we train three random forests
using the sounds that make up the names of Japanese, Chinese, and Korean Pokémon to
classify Pokémon into pre-evolution and post-evolution categories. We then train a fourth
random forest using the results of an elicitation experiment whereby Japanese participants
named previously unseen Pokémon. In Experiment 2, we reproduce those random forests
with name length as a feature and compare the performance of the random forests against
humans in a classification experiment whereby Japanese participants classified the names
elicited in Experiment 1 into pre-and post-evolution categories. Experiment 2 reveals an
issue pertaining to overfitting in Experiment 1 which we resolve using a novel cross-valida-
tion method. The results show that the random forests are efficient learners of systematic
sound-meaning correspondence patterns and can classify samples with greater accuracy
than the human participants.

## Introduction

Natural language processing (NLP) is a field of study that combines computational linguistics
and artificial intelligence and is concerned with giving computers the ability to understand
language in much the same way humans can. The present study tests whether an NLP algo-
rithm can classify samples using sound symbolism, which has been a largely overlooked feature
of human language in NLP. While in modern linguistics, the relationship between sound and
meaning is generally assumed to be arbitrary [1], a growing number of studies have revealed
systematic relationships between sounds and meanings, some of which hold cross-linguisti-
cally. For example, speakers of many languages tend to associate words containing [i] with

**Funding:** AK - Grant obtained from Japan Society for the Promotion of Science (Tokyo, JP) GRANT_NUMBER: 20K13055 https://www.jsps.go.jp/english/index.html The funders had no role in study design, data collection and analysis, decision to publish, or preparation of the manuscript.

**Competing interests:** The authors have declared that no competing interests exist.

small objects, while words containing [a] are typically associated with larger objects [2–5]. Humans understand certain sound symbolic associations in infancy and these associations are said to scaffold language development and facilitate word learning [6–9]. It is therefore important for any NLP algorithm to understand sound symbolism if its goal is to understand language in the same way that humans can. This study is concerned with the random forest algorithm (further RF: [10]), which is an ensemble method machine learning algorithm typically applied to classification and regression tasks. It builds upon recent research by Winter and Perlman ([11]; see also [12]), who used RFs to show that there is a systematic sound-symbolic relationship between size and phonemes in English words.

In the following, we construct and test RFs using the fictional names of characters known as Pokémon. Initially released in 1996 as a video game, Pokémon is an incredibly popular mixed-media franchise, particularly in its country of origin, Japan [13]. The present study measures the classification accuracy of RFs against that of Japanese university students. The RFs are trained to classify Pokémon into pre-evolution and post-evolution categories using only the sounds that make up their names. In Experiment 1, three RFs are constructed using the sounds that make up the names of Japanese, Mandarin Chinese (hereafter: Chinese), and South Korean (hereafter: Korean) Pokémon. These RFs are trained using a subset of each dataset and then tested on the remaining data. While all RFs classify Pokémon at a rate better than chance, the Japanese RF was found to perform the best, hence the remaining experiments are conducted on Japanese participants and Japanese Pokémon names only. The Japanese RF is then tested using the results of an elicitation experiment where Japanese participants were asked to name previously unseen Pokémon presented next to a pre/post-evolution parallel. A further RF is constructed using the responses from the elicitation experiment and tested both on the elicitation responses and the official Japanese names. In Experiment 2, we retrain the RFs presented in Experiment 1 to include name length. These retrained RFs uncover an issue of overfitting caused by a lack of variability in decision trees. We resolve this issue through cross-validation by constructing multiple random forests (MRFs) with different starting values for the randomization of splitting the data into training and testing subsets. The mean accuracy of the RFs in the Japanese MRF is then compared to the results of a classification experiment where Japanese participants were asked to classify the elicited responses from Experiment 1 into pre- and post-evolution categories. The results of the human participants in the categorization experiment are then measured against the results of the MRFs. To summarize, Experiment 1 tests whether RFs can learn to make classification decisions using the sounds that make up names and whether this learning is applicable to elicited samples, and Experiment 2 measures the performance of MRFs against humans.

## Sound symbolism

One of the standard assumptions of modern linguistic theory is that the relationship between sound and meaning is arbitrary [1,14]. While language is undoubtedly capable of associating sounds and meanings in arbitrary ways, the last few decades have seen a growing number of studies that reveal systematic relationships between sounds and meanings [15–17]. One well-known example is the *takete-maluma* effect [18] which is the observation that voiceless obstruents are typically associated with jagged-shaped objects, while names with sonorant sounds are more often associated with round-shaped objects. This effect has been shown to hold cross-linguistically [19–23]. While relationships between sound and meaning can be systematic, they are typically stochastic in nature [24]; that is, sound-meaning relationships manifest themselves as a probability distribution that show statistical skews but may not be hold in all lexical items. For example, English adjectives like *tiny*, *mini*, and *itsy bitsy* adhere to the high front

vowel equates to smallness pattern discussed above, while the English adjective *small* is a clear exception to this generalization [11]. Sound symbolism is demonstrably important for language acquisition processes [21,25]; symbolic words are more common in both child-directed speech and early infant speech [26,27], and indeed, research has shown that infants are sensitive to sound symbolism [6–9].

Pokémonastics is a relatively new subfield of sound symbolism that examines sound symbolic relationships between the names of video game characters known as Pokémon and their attributes. In the video games, the player character collects Pokémon, which they use to battle other players. As Pokémon earn experience, many have the option to evolve. Pokémon evolution permanently changes the Pokémon, they typically grow larger and stronger, and their names change. Pokémonastic studies have shown that Pokémon evolution status can be signaled via some sound symbolic means in English and Japanese by an increase in name length, increased use of voiced obstruents, and in vowel use where the high front vowel [i] is typically associated with pre-evolution Pokémon [28–31]. Based on these established relationships and the likelihood that the participants would be familiar with the subject, Pokémon evolution was determined to be a suitable test case for measuring the ability of RFs against humans in understanding sound symbolism (see also [11]).

## Random forests

RFs, first introduced by Breiman [10], are ensemble method machine learning algorithms that are typically applied to classification and regression tasks. Since their inception, RFs have been a popular tool in machine learning, and several recent review articles attest to their efficacy [32–34]. Typically, RFs work by constructing many decision trees using a two-thirds subset of the data, they are then tested on the remaining data. Decision trees themselves are non-parametric supervised machine learning algorithms that resemble flow charts where each internal node represents a test of features. The decision tree splits at each node based on how important each feature is in the task. Splits eventually lead to a terminal node in the decision tree, which depicts the outcome of the decision-making process. Decision trees can be extremely useful; they are scale-invariant, robust to irrelevant features and inherently interpretable. However, decision trees are sensitive to noise and outliers, and are thus prone to overfitting data which limits their ability to generalize to unobserved samples [35,36]. Overfitting is a modelling error that occurs when a function is too closely aligned to a limited set of data points. This results in a model that performs well for the trained dataset but may not generalize well to other datasets. To address the issue of overfitting, RFs use bootstrap aggregating (bagging: [37]) and the random subspace method [38]. Bagging involves using many decision trees to improve the stability and accuracy of the algorithm by averaging voting (in classification) or the output (in regression). In bagging, samples are randomly allocated to trees, typically with replacement, which raises the issue of duplication. The random subspace method resolves this issue by randomly selecting a subset of features at each internal node, which allows the model to better generalize by introducing variability into the decision trees. In other words, bagging randomly selects samples while the subspace method randomly selects features. By randomizing the decision trees across both dimensions, random forests resolve the issue of overfitting inherent in decision trees.

## Experiment 1: Elicitation

### Material and methods

The data, an explanation of the data, and a detailed annotated script for the following algorithms are available under the OSF repository.

**Official Pokémon name data.** All data were obtained from Bulbapedia ([39], last accessed in June 2022). As of June 2022, Bulbapedia has completed (mainspaced in the parlance of the website) lists for Japanese, Chinese, Korean, English, German, and French Pokémon. Japanese, Chinese, and Korean names were selected for this experiment on the basis that Japanese katakana, Chinese pinyin, and Korean McCune-Reischauer romanisation are reasonably phonetic scripts. An algorithm was created for each language to count the number of times each sound occurs in each name. The algorithms and a detailed explanation for their implementation are included in the above OSF repository. This resulted in an almost entirely phonemic analysis except in the case of tones in Chinese, which are counted as separate features, and voicing on plosives in Korean. In Korean [40] and Chinese [41], there is no phonological opposition between voiced and voiceless plosives. However, Korean plosives are systematically voiced when they occur intervocalically [40], and this is reflected in the McCune-Reischauer romanisation of Korean. Given that voiced plosives have been shown carry information pertaining to Pokémon evolution in other languages [29,31], intervocalic plosives were counted separately in Korean.

As of June 2022, there are 905 Pokémon that span eight generations. This study only examines the names of pre-evolution and post-evolution Pokémon. Some Pokémon do not evolve and are therefore not included in the current study. The sixth generation of the core video game series saw the introduction of a mechanic known as *Mega Evolution* that temporarily transforms certain Pokémon. Mega evolution is not considered by the present study because this is a temporary transformation that has little effect on Pokémon names other than the addition of prefixes like *mega*. Other Pokémon that were excluded from the analysis are mid-stage evolutionary variants. An example of a mid-stage Pokémon is *Electabuzz* which was introduced in the first generation of the video game series. Its pre-evolution variant, *Elekid*, was introduced in the second generation, and its post-evolution variant, *Electivire*, was introduced in the fourth generation. In the present study, we exclude *Electabuzz* from the analysis because it is considered the mid-stage variant, despite other Pokémon being added to the evolutionary family retroactively. Kawahara and Kumagai [28] analysed the relationships between the sounds in the names of Pokémon and Pokémon evolution where they did not exclude mid-stage Pokémon. To achieve this, they had four categories based on evolution level rather than binary pre- and post-evolution categories. RFs are capable of multiclass classification; however, we opted for binary classification for the current analysis because, while the data is technically count data, it is almost entirely binary (e.g., 96.7% of all data points in the Japanese dataset are either 0 or 1). Therefore, it made sense to use a binary classifier given that the sound symbolic patterns are likely scalar across mid- and final-stage categories. The removal of mid-stage Pokémon and Pokémon with no evolutionary family resulted in 628 unique Pokémon names, 303 of which are pre-evolution and 325 of which are post-evolution. The reason for the distribution skew is because certain pre-evolution Pokémon may evolve into multiple post-evolution variants.

**Elicitation experiment.** This experiment received ethics approval from the Nagoya University of Business and Commerce. ID number 21048.

The elicitation experiment has two main goals. The first is to determine whether an RF constructed using the official Pokémon name data can be used to classify names elicited from participants and vice versa. In other words, is there enough overlap between the official names and names provided by participants for each model to be useful in classifying Pokémon from the alternate dataset. The second goal is to provide stimuli for a categorization experiment (Experiment 2) designed to measure the performance of human participants against the machine learning algorithms. To get a fair measurement of classification accuracy, it was

important to test both humans and the machine learning algorithms on data that they had not previously been exposed to, hence the need for elicited samples.

The elicitation experiment was conducted using Google Forms. Each Google form consisted of a short instructional paragraph, followed by twenty Pokémon-like images. Following the method outlined in Kawahara & Kumagai [28], these images were not of existing Pokémon and had likely not been previously viewed by the participants. The instructions noted that only native Japanese speakers were to take the survey. Participants were informed that they were to name twenty new Pokémon. It was made clear to participants that they would be shown images of pre- and post-evolution Pokémon. Participants were asked to provide names for Pokémon in katakana which is the script used for Pokémon names and nonce words in Japanese. Participants were instructed not to use existing words (Japanese or otherwise) to name the Pokémon. Participants were given no further instructions (such as length limitations) regarding naming the Pokémon. Participants were not asked if they were familiar with the Pokémon franchise prior to completing the survey. All instructions were written in Japanese. Participants were informed that their participation was entirely voluntary, that they may quit the survey at any time. Consent was obtained verbally and it was explained to participants that their participation also constituted consent. No personal data were collected other than student email addresses which were collected to ensure that students were not completing the survey twice. These were discarded prior to the analysis.

Each image contained a pre-evolution and a post-evolution Pokémon presented side by side. The pre-evolution Pokémon was always located to the left of the post-evolution Pokémon and was always presented as substantially smaller (see Fig 1) than its post-evolution counterpart. In each image, there was an arrow pointing to the Pokémon that was to be named. Images with arrows pointing to the pre-evolution Pokémon were always followed by an identical image, except the arrow would be pointing to the post-evolution Pokémon. Trials were not randomized, and the pre-evolution image was always followed by the post-evolution image. Pre-evolution Pokémon were always presented on the left and post-evolution Pokemon were always presented on the right. The images were created by a semi-professional artist (DeviantArt user: Involuntary-Twitch), and samples are presented in Fig 1. The images very closely resemble the pixelated images used to represent Pokémon in the earlier generations of Pokémon games.

Participants were recruited from the Nagoya University of Commerce and Business via a post on the student bulletin board. Students were not compensated for their time monetarily or otherwise. The human participants needed to be somewhat familiar with the subject matter because sound-symbolic relationships in fictional names may not adhere to those found in natural languages. Given the popularity of Pokémon in Japan and that the participants were Japanese university students, Pokémon was determined to be a good test case for assessing the accuracy of RFs against that of humans. Forty-nine students responded to the survey. In total, 980 responses were recorded; however, some responses were blank and other responses contained duplicate names, the distribution of which suggested that participants had possibly conferred while taking the survey. These were discarded, resulting in 967 unique names (482 pre-evolution; 485 post-evolution). Elicited names were transcribed using the same algorithm used for the official Japanese Pokémon names. None of the names collected in the elicitation experiment were names of existing Pokémon.

**Random forests.**   Random forests were constructed and tested using the ranger package 0.13.1 [42]. The number of trees included in each RF was manually tuned by constructing nine RFs at different tree number values with different starting points for randomization (set.seed). Optimal values were determined by examining mean out of bag (OOB) accuracy and its standard deviation. OOB error refers to incorrectly classified samples. For all RFs, 20,000 trees

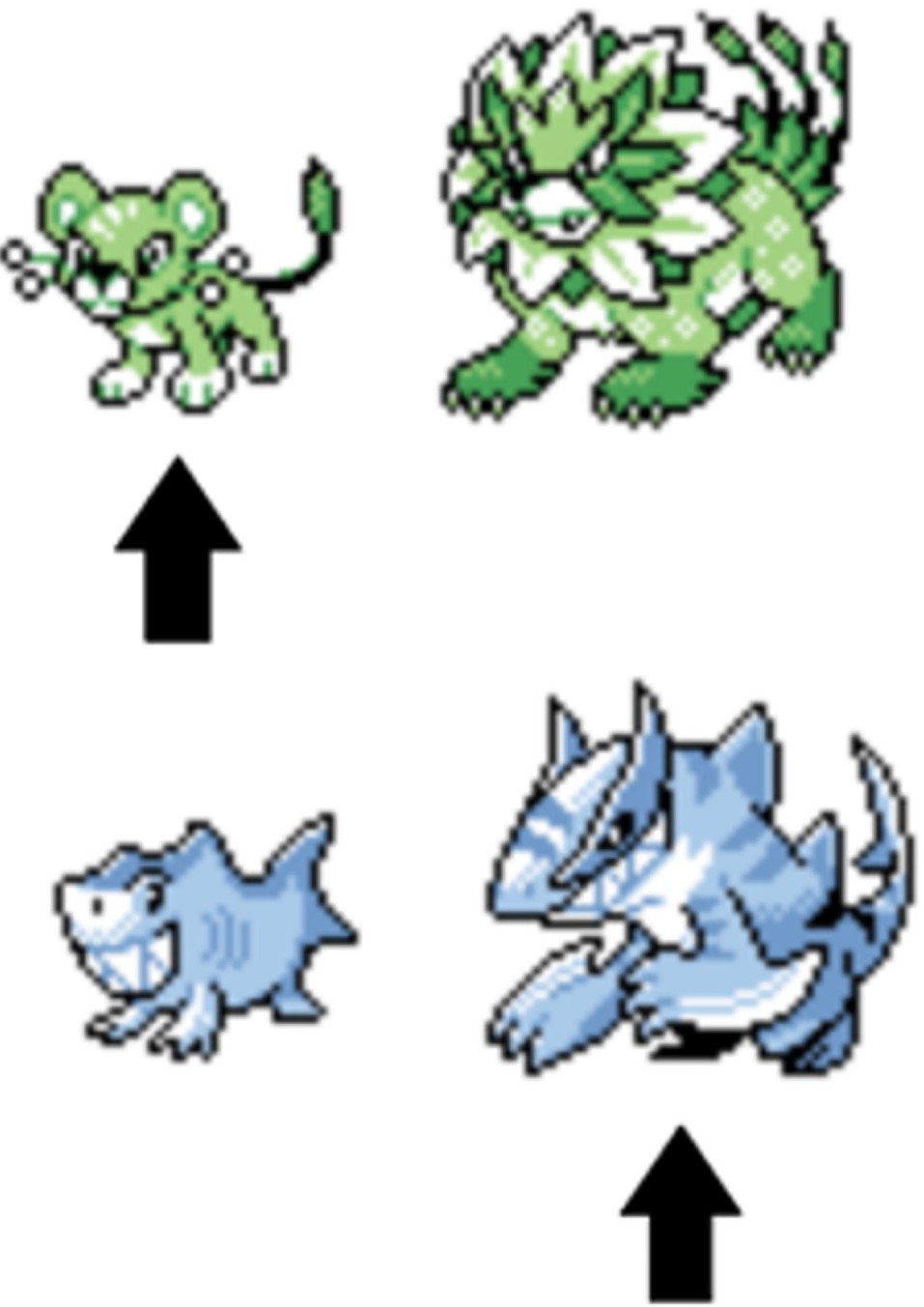

**Fig 1. Sample stimulus pairs of pre- and post-evolution Pokémon characters used in Experiment 1.** These images are reproduced with the permission of the artist.

were determined to be a suitable size because we observed no reduction in OOB error with increased trees and because calculating feature importance using the Altmann method [43] at 20,000 trees approached the processing capability of the computer the RFs were constructed upon. Hyperparameters pertaining to the number of features examined at each node, the sample fraction, and node size were tuned using the tuneRanger package 0.5 [44]. Essentially, the tuning process determines how much variability there is between trees. Highly variable trees

will produce highly variable results but might encounter issues with datasets that contain many unimportant features or null values. Low variability in decision trees results in more stable algorithms but may mask the importance of weaker features because they will often be paired with strong features. The accuracy of the RF is determined by feeding the testing data into the model and assessing the OOB error. The OOB error gives an overall representation of the accuracy of the algorithm but does not communicate which features are important in classification, which is instead determined by feature importance. There are several ways to calculate feature importance, the present study uses permutation. In permutation, each feature is randomized individually, and then the algorithm is reconstructed with all other features remaining the same. Feature importance is calculated on the increase of OOB error due to randomization. One issue with the interpretability of RFs is that feature importance does not communicate directionality. For example, those sounds that are important to classification may be considered as "pulling" each sample into one category or the other, while feature importance communicates the strength of the "pull", it does not communicate whether that "pull" is in the direction of the pre- or post-evolution category. In the present study, we report on the distribution of speech sounds to pre- and post-evolution categories.to indicate directionality, though it should be noted that they are not necessarily the same measure.

In total, there were six RFs constructed for Experiment 1. The first three RFs presented in the results section were trained using a randomly sampled subset consisting of two-thirds of the Japanese, Chinese and Korean Pokémon names. The fourth RF is trained using two-thirds of the results of the elicitation experiment. All four RFs are then tested using the remaining one-third subset of each dataset. We then calculate feature importance for each RF to examine potential cross-linguistic patterns, and patterns between the Japanese Pokémon data and the elicited data. The remaining two RFs are constructed using the entirety of the official Japanese Pokémon names and the entirety of the samples collected in the Elicitation experiment. These two RFs are then tested using the alternate dataset. In other words, one RF is constructed using all the official names and tested on the elicited responses, while the other is constructed using all the elicited samples and tested on the official names. This is done to determine whether there is enough overlap in the two datasets for the algorithms to be useful in classifying the opposite dataset.

## Results

The three RFs trained and tested on the official Pokémon names all classified Pokémon at a rate better than chance. Given that there is an uneven distribution of pre- and post-evolution Pokémon, any model that naïvely classified to the majority category would achieve an accuracy of around 52% (OOB error 48%) depending on the split of the training and testing subsets. The Japanese RF was the most accurate (OOB error 29.05%), followed by the Chinese RF (OOB error 39.05%), and finally, the Korean RF (OOB error 40.95%). A confusion matrix for the Japanese RF is presented in Table 1 and feature importance for the Japanese RF is presented in Table 2. Note here that in Experiment 2, we report on the results of MRFs with different starting values for the randomization of both splitting the data in the training and testing

**Table 1. Confusion matrix for the Japanese RF.**

|  |  | Classification | |
| --- | --- | --- | --- |
|  |  | Pre-evolution | Post-evolution |
| Sample | Pre-evolution | 69 | 38 |
|  | Post-evolution | 23 | 80 |

**Table 2. Feature importance (Importance) and *p* values for features that achieved a feature importance greater than 0.1% in the Japanese RF.**

| Feature | Importance | *p* value |
|---|---|---|
| /m/ | 0.78% | 0.030 |
| /ɴ/[a] | 0.63% | 0.020 |
| /:/[b] | 0.45% | 0.049 |
| /g/ | 0.40% | 0.049 |
| /a/ | 0.39% | 0.139 |
| /ɾ/ | 0.35% | 0.089 |
| /Q/[c] | 0.24% | 0.059 |
| /ɸ/ | 0.20% | 0.079 |
| /t͡ɕ/ | 0.19% | 0.069 |
| /d͡ʒ/ | 0.19% | 0.129 |
| /d/ | 0.19% | 0.228 |

[a] /ɴ/ represents the coda nasal.

[b] /:/ represents the second portion of long vowels.

[c] /Q/ represents the initial portion of geminate consonants.

subsets, and the RFs themselves. The results of the MRFs (OOB error: M = 34.07%, SD = 2.48%) suggest that this result was an outlier caused by a particularly advantageous split between training and testing subsets. This process was conducted for the Chinese (OOB error: M = 40.85%, SD = 3.35%) and Korean (OOB error: M = 43.28%, SD = 3.09%) datasets as well. The RF trained and tested on the elicited names (Elicited RF) classified samples at a rate better than chance. As with the official datasets, there was an uneven distribution to categories, a naïve model would accurately classify samples in the elicited data 50.16% (OOB error 49.84%) of the time. The Elicited RF achieved an OOB error of 30.96%. Feature importance was calculated for each model to determine which sounds contributed to classification. Feature importance and significance is calculated using the Altmann [43] permutation method on the training subsets. Permutation involves randomizing features individually; the random forest is then reconstructed for each feature. Feature importance is the increase in OOB error for the feature being randomized. The Altmann permutation method involves running multiple permutations to estimate more precise *p* values. Feature importance significance is calculated by normalizing the biased measure based on a permutation test. This returns a significance result for each feature, not for the random forest itself [43]. All RFs in the present study use the Altmann permutation method with the number of iterations set at 100. Directionality was determined by the distribution of features in the training subsets of the data. The distribution of features in the Japanese training subset is presented in Fig 2. In the Japanese RF, the most important features were the bilabial nasal (/m/), the coda nasal (/ɴ/), long vowels (/:/), and the voiced velar plosive (/g/). Of these features, only /m/ occurs more frequently in the pre-evolution samples.

As with the Japanese RF, the distribution of most features that were important in the Chinese RF skewed towards the post-evolution category. A confusion matrix for the Chinese RF is presented in Table 3 and feature importance scores for its features are presented in Table 4, and distribution is presented in Fig 3. Tones are an important feature in the RF; where the falling tone occurs more frequently in the post-evolution samples, the neutral tone occurs more frequently in the pre-evolution samples. The velar nasal (/ŋ/) was also found to be an important feature in the Chinese RF.

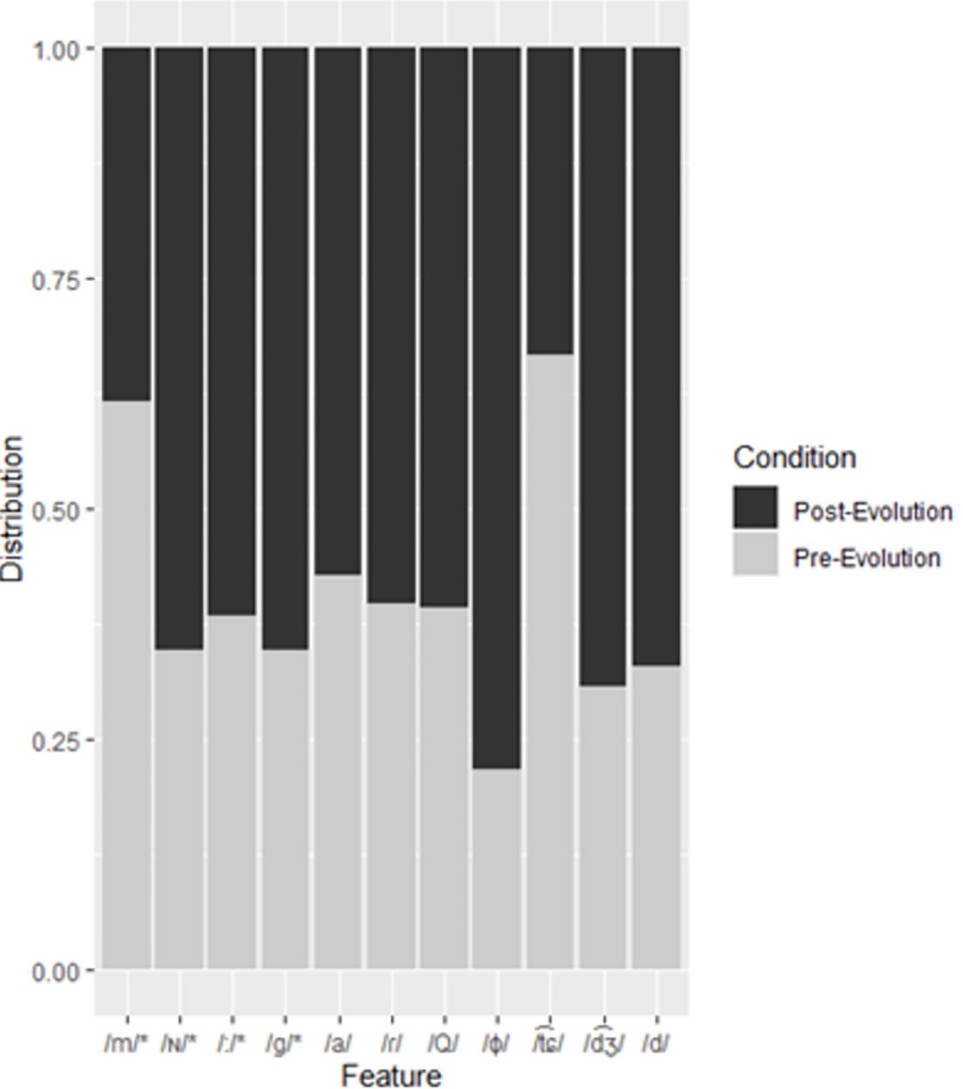

**Fig 2. Distribution of features to pre- and post-evolution categories in the Japanese training subset.** Features appear in order of importance from left to right. Asterisks denote significant features.

In the Korean RF, vowels /ɯ/, /a/, /ʌ/, and /u/ were important, as was the voiced labial-velar approximant /w/. Interestingly, while the close back unrounded vowel, /ɯ/ was present more often in post-evolution samples, the close back rounded vowel /u/ was present more often in pre-evolution samples. Table 5 presents a confusion matrix for the Korean RF, Table 6 presents the feature importance and p values, and Fig 4 presents the distribution.

**Table 3. Feature importance (Importance) and *p* values for features that achieved a feature importance greater than 0.1% in the Chinese RF.**

| | | Classification | |
|---|---|---|---|
| | | **Pre-evolution** | **Post-evolution** |
| Sample | Pre-evolution | 53 | 50 |
| | Post-evolution | 29 | 78 |

**Table 4. Feature importance (Importance) and *p* values for features that achieved a feature importance greater than 0.1% in the Chinese RF.**

| Feature | Importance | *p* value |
|---|---|---|
| Falling tone | 0.88% | 0.020 |
| /ŋ/ | 0.87% | 0.010 |
| /t͡ɕ/ | 0.20% | 0.089 |
| /ɕ/ | 0.14% | 0.109 |
| /e/ | 0.13% | 0.238 |
| /o/ | 0.13% | 0.257 |
| Neutral tone | 0.13% | 0.188 |

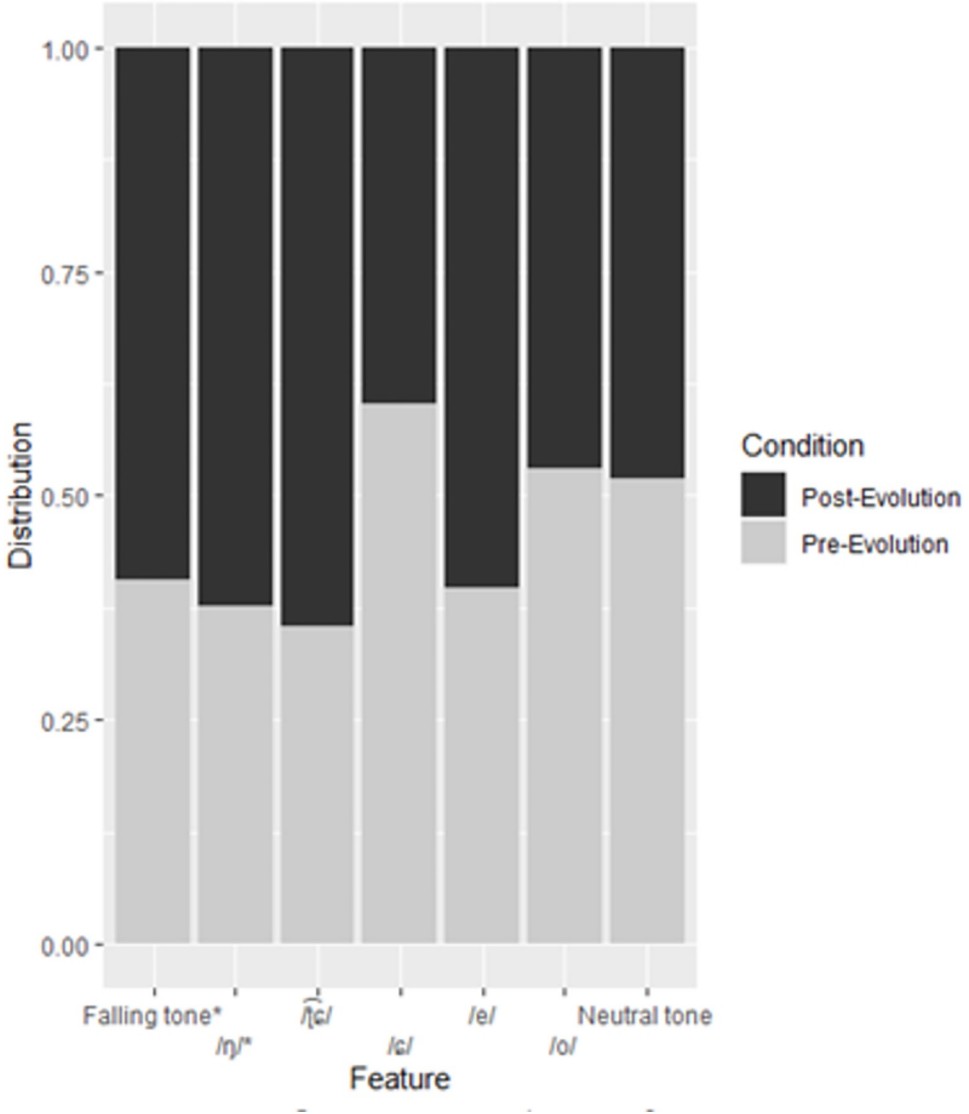

**Fig 3. Distribution of features to pre- and post-evolution categories in the Chinese training subset.** Features appear in order of importance from left to right. Asterisks denote significant features.

**Table 5. Confusion matrix for the Korean RF.**

| | | Classification | |
|---|---|---|---|
| | | Pre-evolution | Post-evolution |
| Sample | Pre-evolution | 57 | 50 |
| | Post-evolution | 36 | 67 |

**Table 6. Feature importance (Imp.) and p values (p) for features that achieved a feature importance greater than 0.1% in the Korean RF.**

| Feature | Importance | p value |
|---|---|---|
| /ɯ/ | 2.558% | <0.001 |
| /a/ | 1.326% | 0.0297 |
| /w/ | 0.226% | 0.0693 |
| /ʌ/ | 0.207% | 0.1287 |
| /u/ | 0.113% | 0.1782 |

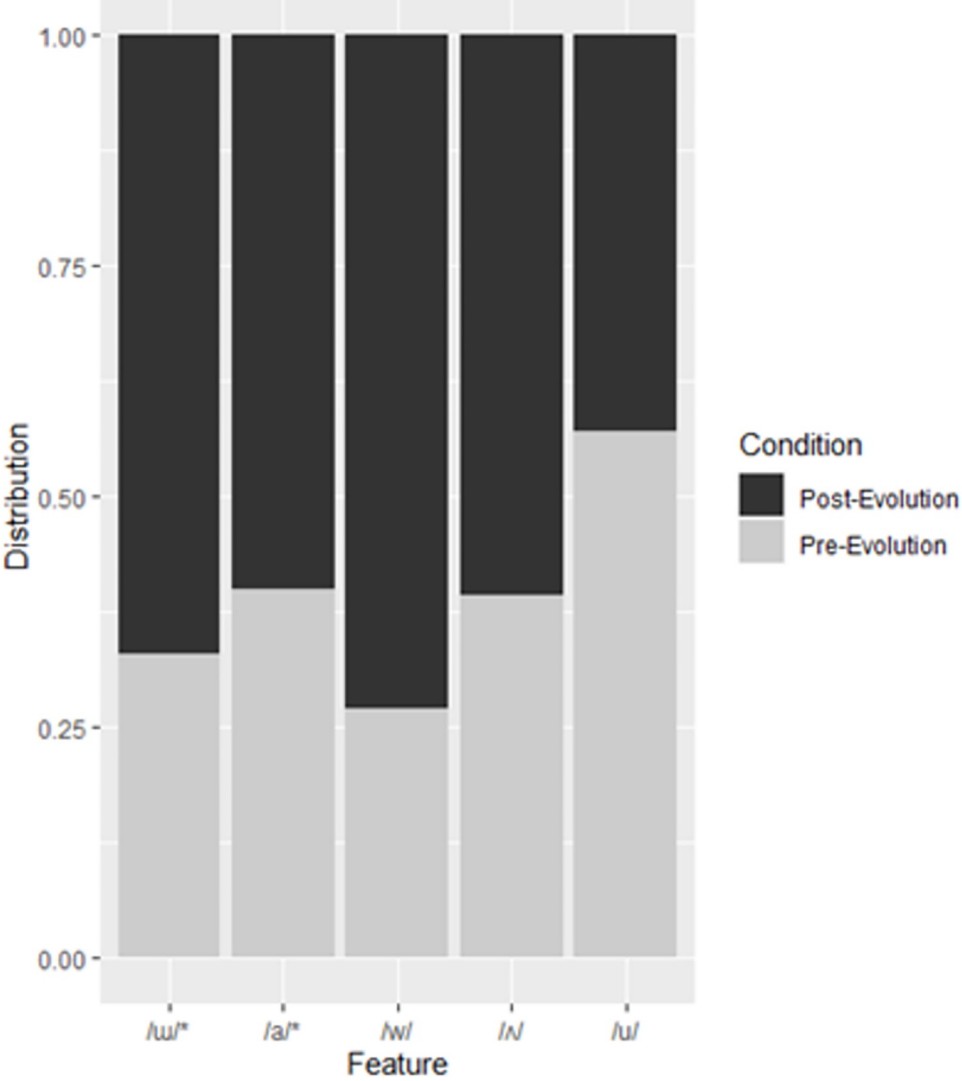

**Fig 4. Distribution of features to pre- and post-evolution categories in the Korean training subset.** Features appear in order of importance from left to right. Asterisks denote significant features.

**Table 7. Confusion matrix for the Elicited RF.**

| | | Classification | |
|---|---|---|---|
| | | Pre-evolution | Post-evolution |
| Sample | Pre-evolution | 103 | 55 |
| | Post-evolution | 45 | 120 |

**Table 8. Feature importance (Imp.) and p values (p) for features that achieved a feature importance greater than 0.1% in the Elicited RF.**

| Feature | Importance | *p* value |
|---|---|---|
| /d/ | 1.263% | <0.001 |
| /ɯ/ | 1.059% | <0.001 |
| /a/ | 0.918% | <0.001 |
| /g/ | 0.838% | <0.001 |
| /d͡ʒ/ | 0.448% | <0.001 |
| /o/ | 0.219% | 0.2277 |
| /ɴ/ | 0.180% | 0.2376 |
| /:/ | 0.140% | 0.2772 |
| /z/ | 0.138% | 0.1287 |

Most of the features that were important in the Japanese RF were also important in the Elicited RF. These include voiced plosives (/g/ & /d/), the open front unrounded vowel (/a/), the coda nasal (/ɴ/), and long vowels (/:/). Interestingly, all the features that achieved a feature importance greater than 0.1% in the Elicited RF occurred more frequently in post-evolution Pokémon. The confusion matrix for the RF constructed and tested on the data from the elicitation experiment are presented in Tables 7 and 8 presents the feature importance scores, and Fig 5 presents the distribution chart.

Given that the Japanese RF and the Elicited RF feature importance patterns are reasonably similar, we wanted to test whether these RFs would be able to accurately classify samples from their opposite dataset. Important features that are shared between the two models are non-labial voiced obstruents such as /d/ and /g/, coda nasals, long vowels, and the low front vowel /a/. Interestingly, the distributional skew for all of these features is towards the post-evolution category. We tested each existing RF on the entirety of their opposite dataset (not just the test subsets). The Japanese RF was able to accurately classify the elicited samples 61.43% of the time (OOB error 38.57%), and the Elicited RF was able to accurately classify the official Japanese Pokémon name samples 66.72% of the time (OOB error 33.28%) where naïve models would be expected to achieve an accuracy of 52% and 50.16% respectively. The confusion matrix for the RF trained using the official Japanese Pokémon names and tested using the elicited samples is shown in Table 9. Table 10 shows the confusion matrix for the for the RF trained using the elicited samples and tested using the official Japanese Pokémon names.

## Discussion

All the RFs presented above performed better than a naïve algorithm would. For the Japanese, Chinese, and Korean RFs, a naïve algorithm would be expected to achieve an OOB error of 48%. While the Japanese RF was shown to be the most accurate (OOB error 29.05%), the Chinese (OOB error 39.05%) and Korean (OOB error 40.95%) error rates were well below 48%. The elicited RF, for which a naïve algorithm would be expected to achieve an OOB error of 50%, achieved an OOB error of 30.96%. Important to remember here is that the RFs were

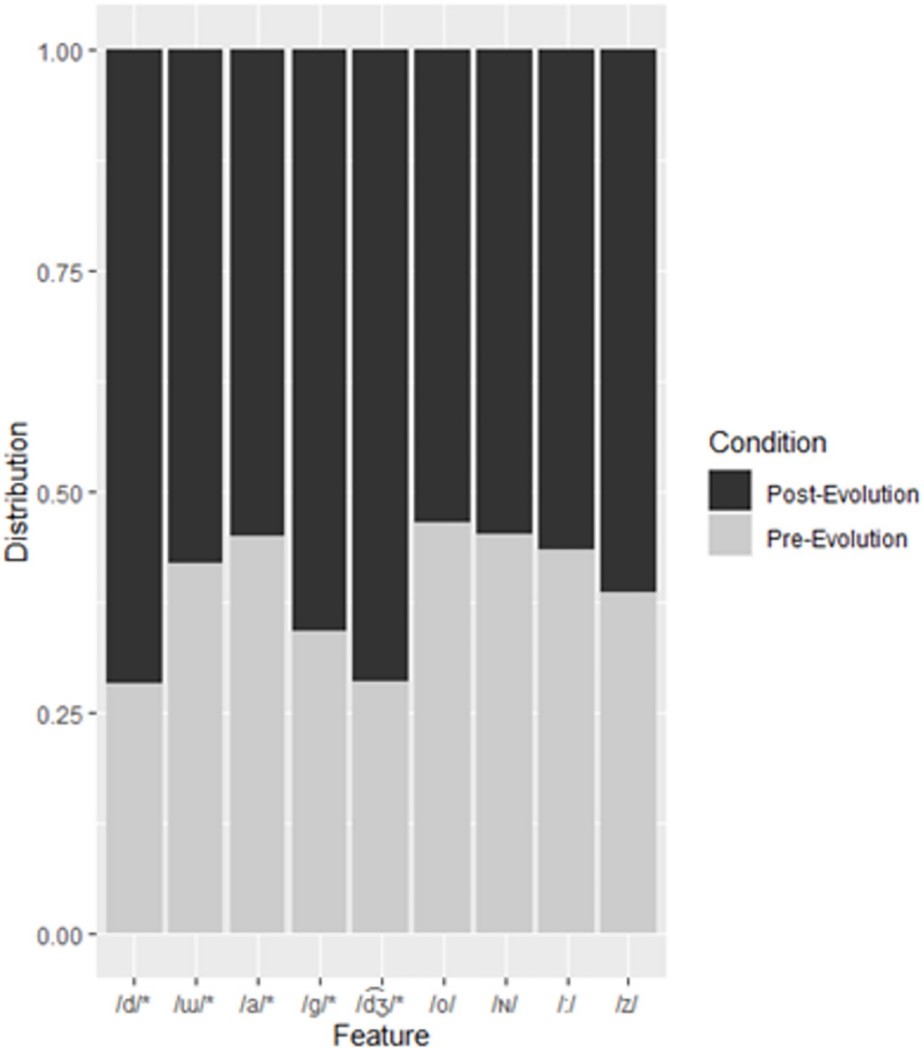

**Fig 5. Distribution of features to pre- and post-evolution categories in the elicited training subset.** Features appear in order of importance from left to right. Asterisks denote significant features.

trained on only two-thirds of the data, so the RFs were efficient learners, given that they only had 419 samples to learn from. The RF trained on the official Japanese data and tested on the elicited data was trained on all 628 official Japanese samples and tested on all 967 elicited responses. Despite having more samples from which to learn, the RF trained on the official names and tested on the elicited responses (OOB error 38.57%) was less accurate than the RF trained and tested on the official names (OOB error 29.05%). Similarly, the RF trained on the

**Table 9. Confusion matrix for the RF trained using the official Japanese Pokémon names and tested using the elicited samples.**

|  |  | Classification | |
|---|---|---|---|
|  |  | **Pre-evolution** | **Post-evolution** |
| Sample | Pre-evolution | 245 | 237 |
|  | Post-evolution | 136 | 349 |

**Table 10. Confusion matrix for the for the RF trained using the elicited samples and tested using the official Japanese Pokémon names.**

| | | Classification | |
|---|---|---|---|
| | | **Pre-evolution** | **Post-evolution** |
| Sample | Pre-evolution | 180 | 123 |
| | Post-evolution | 83 | 242 |

entirety of the elicited responses and tested on the official names (OOB error 33.28%) was less accurate than the RF trained and tested on the elicited responses (OOB error 30.96%). Despite these differences, the official names and the elicited responses are similar enough to perform better than naïve algorithms.

The feature importance scores of the RFs reveal interesting relationships between Pokémon evolution status and the sounds that make up their names, some of which hold across languages. While high front vowels did not achieve a feature importance greater than 0.1% in any of the RFs, the low front vowel /a/ and the high back unrounded vowel /ɯ/ were important in the Japanese, Korean, and Elicited RFs and were distributionally skewed towards post-evolution in all cases. The result for the phoneme /a/ as representing post-evolution Pokémon is in line with the well-known observation that nonce words containing [a] are larger than those containing [i] [2,45,46] given that post-evolution Pokémon are typically larger than their pre-evolution counterparts. Interestingly, the high back rounded vowel /u/ was important in the Chinese model, but it skewed towards the pre-evolution category. Vowels were found to be important in the Korean model, particularly /ɯ/, /a/, /ʌ/, and /u/. Korean vowels have been found to hold sound symbolic correspondences between "light" and "dark" vowels [47]. These correspondences run counter to cross-linguistic patterns. For example, light vowels are defined as being low vowels and are said to reflect small, fast-moving entities, while dark (or high) vowels are said to reflect larger, slow-moving entities [48]. Our findings do not support this observation. Although the distribution of dark vowels /ɯ/ and /ʌ/ skew towards the post-evolution category, the distribution of the light vowel /a/ skews towards the post-evolution category, while the dark vowel /u/ skews towards the pre-evolution category. The finding that /a/ is important to the Korean model and skews towards the post-evolution category is in line with [5] who found that Korean listeners judge nonce words to be larger when the contain [a]. Long vowels were important in both the Japanese and the Elicited RFs, and they skewed towards post-evolution in both cases. This finding is reflected in previous Pokemon studies [29,30], which also show that long vowels are associated with increased size. These studies tend to suggest this can be explained by the *iconicity of quantity* which is the finding that larger objects are typically associated with longer names [49]. This is explored further in Experiment 2. Lastly, tones in the Chinese RF were important to the model. The falling tone had the highest feature importance in the Chinese RF and it skewed toward the post-evolution category. The neutral tone, on the other hand, skewed toward the pre-evolution category. In a similar Pokémonastic study, Shih et al., [50] found that the falling tone seems to be associated with increased power, evolution stage, and increased distribution to the male gender. This is seemingly more complex than what Ohala's Frequency Code hypothesis [51] would predict as it simply states that low tones should reflect largeness while high tones should predict smallness; but makes no prediction regarding tone pitch contour. Shih et al., [50] propose that the falling tone has the steepest pitch of all Chinese tones, and that this may explain why this tone is iconically linked to largeness in Chinese.

The Japanese nasal /ɴ/ and the Chinese nasal /ŋ/ were important in the Japanese, Chinese, and Elicited RFs and skewed towards post-evolution in all cases. This is an interesting finding

given that both consonants can only occur in the coda position, although the coda nasal /ŋ/ in Korean did not achieve a feature importance greater than 0.1%. Cross-linguistically, nasal consonants are generally associated with large entities [2,52], likely due to their low frequency [2]. In Japanese, however, bilabial consonants have been found to be associated with images of cuteness and softness [53], which may explain why /m/ was both important in the Japanese model and was skewed towards the pre-evolution category. High back vowels in the Korean model present an interesting case study when examined through the lens of the relationship between cuteness and labiality in Japanese. In the Korean model, both high back vowels were found to be important. While the high back rounded vowel /u/ skewed towards the pre-evolution category, the high back unrounded vowel /ɯ/ skewed towards the post-evolution category. This result suggests that the association between cuteness and labiality may be a cross-linguistic one; however, this suggestion is tentative given that the Korean labial-velar approximant both skewed towards post-evolution and was important in the model. Berlin [2] suggests that nasal consonants can imply largeness given their low frequency energy; however, the bilabial nasal /m/ skewed towards the pre-evolution category and was found to be important in the Japanese RF. In line with Shih et al. [31], who found that voiced plosives were reflective of size in Japanese and English Pokémon names, voiced plosives /d/ and /g/ were important in the Japanese and Elicited RFs. Intervocalic plosives in Korean were counted separately due to maintaining systematic voicing in these positions; however, these did not achieve a feature importance greater than 0.1%.

## Experiment 2: Categorization

### Random forests

The RFs presented in Experiment 1 were constructed using only the sounds that make up the names of Pokémon. In Experiment 2, we reconstruct those RFs with name length as an additional feature. Length was not included in the previous RFs because previous studies suggest that it is likely a highly important feature [29,31], the inclusion of which would likely mask the importance of other features. In all other aspects, the RFs presented in Experiment 2 follow the same method as those in Experiment 1, except in the case where multiple random forests (MRFs) are constructed independently of each other. In MRFs, the starting value for the randomization for splitting data into training and testing subsets, as well as the starting value for the randomization for each RF was set as the number where the RF fell in the RF sequence. So the first RF in each MRF had a set.seed value of 1 while the ninth RF had a set.seed value of 9. For MRFs, tuning was conducted on the first RF only and those hyperparameter settings were applied to all nine RFs in each MRF. This is because, as far as we can tell, there is no way to make the tuning process replicable and the data for individual RFs come from the same source. We test each MRF nine times using the testing subset for each random split. In those instances where the entirety of a dataset is tested against the entirety of a different dataset, as in the case of testing the accuracy of the official Japanese MRF against the elicited Japanese MRF, nine iterations of each test were run with different starting values for the randomization of the MRFs.

### Categorization experiment

This experiment received ethics approval from the Nagoya University of Business and Commerce. ID number 21057.

In the categorization experiment, we took the elicited responses from Experiment 1 and asked Japanese participants to classify them as either pre-evolution or post-evolution Pokémon. One hundred samples were selected randomly from the 967 elicited responses. There

was no control for distribution in the random sampling process because there is no distribution control in the splitting of subsets for RFs. These samples were used to populate five Google Forms that held twenty elicited names each. The surveys were designed in this manner, rather than randomly sampling twenty names from the entire dataset, to simulate the voting process that decision trees undertake in RFs. This is discussed further in the results section below. The forms explained (in Japanese) to the participants that they were to assign new Pokémon to either pre- or post-evolution categories. These choices were presented as buttons labelled 進化前 [pre-evolution] and 進化後 [post-evolution]. Participants were not asked if they were familiar with the Pokémon franchise prior to completing the survey. Five QR codes were generated for each of the five Google Forms. The codes were printed on handouts and distributed to Japanese university students at the Aichi Prefectural University and the Nagoya University of Business and Commerce. Other than the QR code, there was no other information on the handout except for the heading ポケモンクイズ [Pokémon Quiz]. Handouts were distributed to students prior to club activities and scheduled classes. Students were not given any time in class to complete the survey. In total, 119 participants responded to the survey, and there were 10 instances where participants had failed to designate a category, resulting in 2,370 responses. As with Experiment 1, participants were informed that their participation was entirely voluntary, that they may quit the survey at any time. Consent was obtained verbally. No personal data were collected other than student email addresses which were collected to ensure that students were not completing the survey twice. These were discarded prior to the analysis. It was requested that students who had undertaken Experiment 1 were to refrain from taking Experiment 2.

## Results

The aim of Experiment 2 is to compare the performance of RFs against that of humans in classifying Pokémon into pre- and post-evolution categories. In the categorization experiment, human participants had access to name length, so length was included in the algorithms to give the RFs access to this information. In the following, the distribution of length is examined across pre-and post-evolution in all four datasets. Then, all previous RFs are reconstructed to include length to ascertain its effects on OOB error. We also calculate the feature importance of length to determine how much it is contributing to OOB error. Finally, we report on the results of the categorization experiment and compare the accuracy of the human participants against that of the machine learning algorithms. Length was calculated on the sum of all sounds in each dataset except for Chinese tones. In an exploration of sound symbolic relationships in Pokémon names, Kawahara and Kumagai [28] calculated name length on the number of moras in Japanese names because the mora is the most psycholinguistically salient prosodic unit [54]. Although decision trees are scale-invariant, we calculated length on the number of features to bring the Japanese length parameter in line with the Chinese and Korean parameters. Chinese tones were excluded from the length calculation because tones are a measure of pitch contour and do not contribute to the overall length of a name the same way that other speech sounds do. Despite this, Chinese names were longer than those in all other data sets, with both pre-evolution ($M = 8.76$, $SD = 2.09$) and post-evolution ($M = 9.31$, $SD = 2.09$) Pokémon names consisting of a median of nine sounds. Length in the Japanese, Korean, and elicited datasets were similar, with all pre-evolution names consisting of a median of seven sounds, and all post-evolution names consisting of a median of eight sounds. The difference between mean pre- and post-evolution length was greatest in the Elicited dataset (1.52), followed by the Japanese dataset (0.9), Korean (0.73), and finally Chinese (0.55). Mean, median and standard deviation for length across datasets are presented in Table 11. Fig 6 presents a boxplot of length in Chinese, Japanese, and Korean by evolution status.

**Table 11. Feature importance (Imp.) and p values (p) for features that achieved a feature importance greater than 0.1% in the Korean RF.**

| Language | Measure | Pre-evolution | Post-evolution |
|---|---|---|---|
| Chinese | Median | 9 | 9 |
| | Mean | 8.76 | 9.31 |
| | Standard Deviation | 2.09 | 2.09 |
| Japanese | Median | 7 | 8 |
| | Mean | 7.28 | 8.18 |
| | Standard Deviation | 1.38 | 1.26 |
| Korean | Median | 7 | 8 |
| | Mean | 7.33 | 8.06 |
| | Standard Deviation | 1.88 | 1.81 |
| Elicited | Median | 7 | 8 |
| | Mean | 6.79 | 8.31 |
| | Standard Deviation | 2.07 | 2.38 |

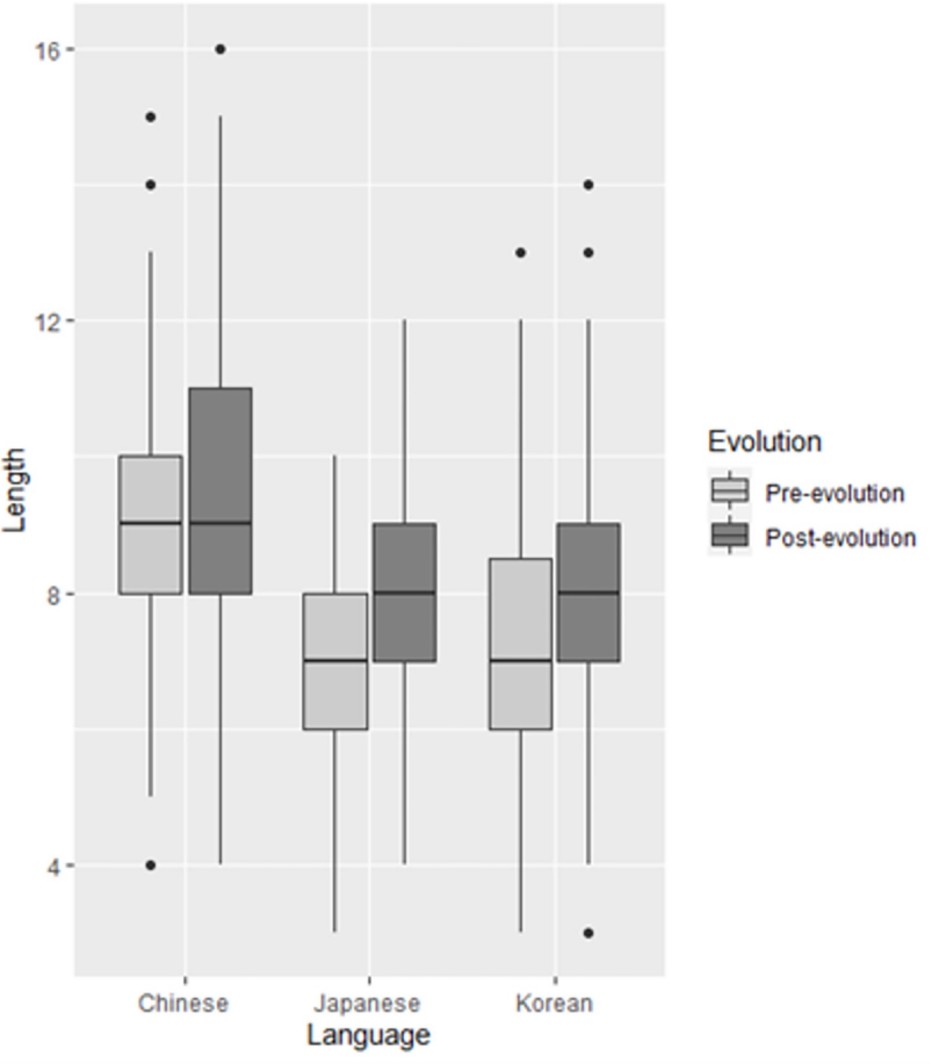

**Fig 6. Boxplot of length for pre- and post-evolution Pokémon across the three languages.**

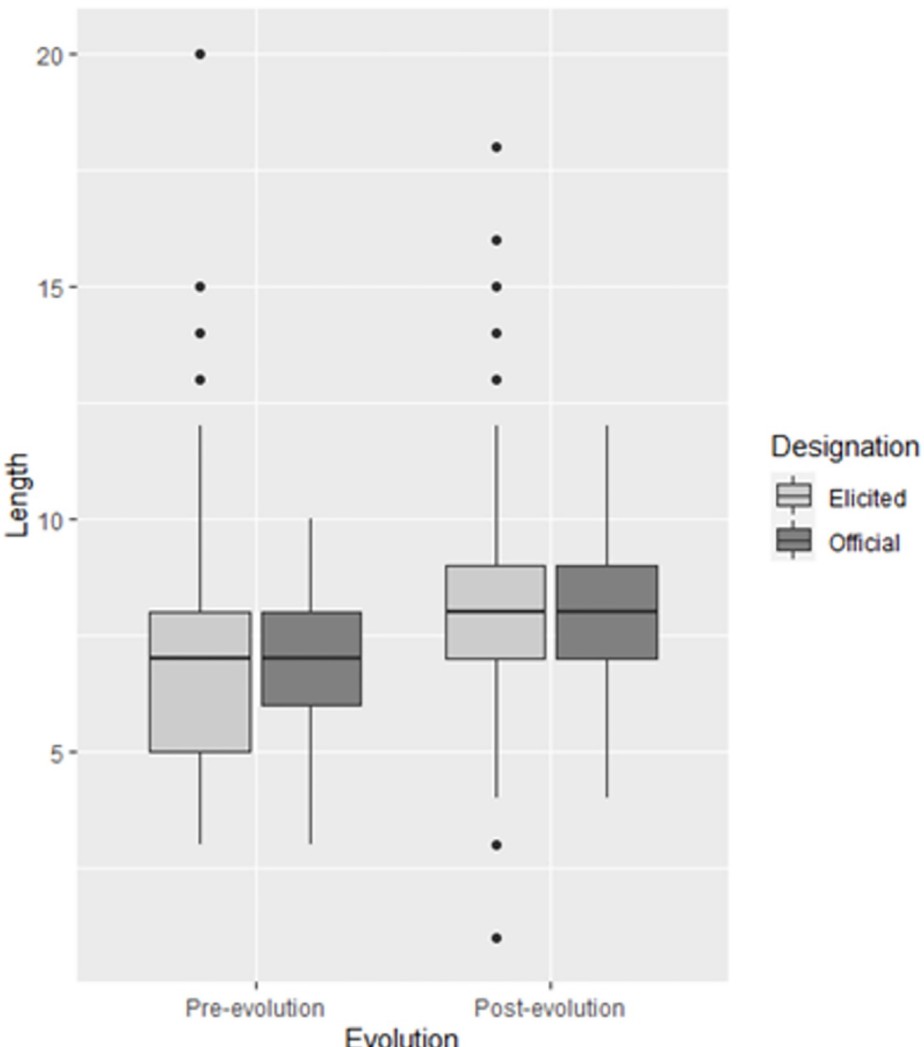

**Fig 7. Official Japanese Pokémon name length (Official) and elicited name length (Elicited) presented side by side in a boxplot.**

The distribution of length in the official Japanese Pokémon dataset and the elicited dataset were extremely similar. While there were more outliers in the elicited dataset, the median, upper and lower quartiles, and minimum and maximum scores (excluding outliers) were almost identical. Fig 7 presents a boxplot for the official Japanese Pokémon name length, and the elicited name length presented side by side to illustrate these similarities.

Length was excluded from the RFs in the previous section because it is clear from previous studies that length would be an important feature [29,31] and would likely mask the importance of speech sounds. Its inclusion should therefore increase the accuracy of the models (or reduce OOB error). However, this was not the case. All previous RFs were reconstructed to include the length feature. These RFs underwent the same tuning process outlined in Experiment 1. The feature importance of length is presented in Table 12. Table 12 also presents the OOB error rates for the RFs constructed with (+L) and without (-L) length. The inclusion of length increased the OOB error of all but the Chinese RF, which should be the RF least affected by length because the difference in average length between pre- and post-evolution Pokémon

**Table 12. OOB error rates for the RFs constructed in Experiment 1 (-L OOB), the OOB error for RFs constructed using length (+L OOB), and the feature importance of length in those RFs (L Imp).**

| RF | -L OOB[a] | +L OOB[b] | L Imp[c] |
|---|---|---|---|
| Chinese | 41.90% | 41.43% | 0.25% |
| Japanese | 29.05% | 30.95% | 4.74% |
| Korean | 40.95% | 41.43% | 1.33% |
| Elicited | 30.34% | 30.96% | 6.66% |

was the smallest in the Chinese dataset. Confoundingly, length was shown to be an important feature in the RFs, yet its effects were not being exhibited by the difference in OOB error between +L and -L RFs.

To explore a potential explanation for this, we examined the randomization processes used in the construction of RFs. For all RFs until this point, we used the same set.seed value, except for those used to tune the number of trees in each forest (num.trees). This value was used as the starting number to generate randomization for both the splits between training and testing data and the RFs themselves. In the method section of Experiment 1, we tuned num.trees by running nine iterations of each num.trees value with different set.seed values for the randomization of RFs. We applied this method to the randomization of the splits between training and testing subsets and found a substantial amount of variation in OOB error. We ran nine iterations of each of the RFs presented in this study. Here, however, we adjusted the set.seed values for both the subset splits and the RFs. The set.seed values ranged from 1–9 for both +L and -L RFs, resulting in the same nine subset splits. The results of these are displayed in Table 13. There is no way to control for randomization in the tuning process, each time the tuning process is conducted, it returns different hyperparameter values even when conducted on the same data. Given that the nature of the data remained the same, the hyperparameter values used for the MRFs were taken from the previous RFs.

To understand the reason why the Japanese RF in Experiment 1 achieved such a low OOB error, we examined the mean feature importance values for features in the -L Japanese MRFs and compared them to the feature importance of features in the -L Japanese RF. Table 14 shows the confusion matrix for the Japanese MRF. Table 15 presents the feature importance values in the Japanese RF and the mean feature importance values in the Japanese MRF. Here we see that the Japanese RF outlined in Experiment 1 was over-emphasising the importance of features /m/, /ɸ/, and /Q/, and under-emphasising the importance of /ɴ/, /ː/, /ɾ/, /d/, /ɯ/, /o/, and /s/. The latter three were not included in earlier tables and charts because they did not achieve a feature importance greater than 0.1% in the Japanese RF.

Given that the randomization of subsets has such a large impact on OOB error, we reconducted the tests of the Japanese RF using the elicited data and the Elicited RF using the official Japanese Pokémon data using both -L and +L datasets. We tested the entirety of the Japanese

**Table 13. Results of multiple random forests.** The mean OOB error for RFs constructed without length (-L OOBM) and their standard deviation (-L OOBSD), the mean OOB error for the RFS constructed with length (+L OOBM) and their standard deviation (+L OOBSD), and the mean feature importance of length (L ImpM) in the +L MRFs.

| MRF | -L OOB[M] | -L OOB[SD] | +L OOB[M] | +L OOB[SD] | L Imp[M] |
|---|---|---|---|---|---|
| Chinese | 40.85% | 3.35% | 39.36% | 4.52% | 0.51% |
| Japanese | 34.07% | 2.40% | 31.69% | 3.01% | 5.47% |
| Korean | 43.28% | 3.09% | 40.85% | 2.85% | 2.44% |
| Elicited | 36.29% | 1.98% | 32.47% | 2.44% | 6.99% |

**Table 14. Confusion matrix for the Japanese MRF trained and tested on multiple subsets from the official Poké-mon names that include length as a feature.**

| | | Classification | |
|---|---|---|---|
| | | Pre-evolution | Post-evolution |
| Sample | Pre-evolution | 609 | 314 |
| | Post-evolution | 285 | 682 |

**Table 15. Feature importance of sounds for Japanese RF trained on official Pokémon names (RF Imp), mean feature importance of sounds for Japanese MRF trained on official names (MRF ImpM), and mean standard deviation for the Japanese MRF (MRF ImpSD).** Asterisks reflect a mean p value of less than 0.05.

| Feature | RF Imp | MRF Imp$^M$ | MRF Imp$^{SD}$ |
|---|---|---|---|
| /m/ | 0.78%* | 0.39%* | 0.22% |
| /ɴ/ | 0.63%* | 1.29%* | 0.38% |
| /:/ | 0.45%* | 0.83%* | 0.21% |
| /g/ | 0.40%* | 0.45% | 0.20% |
| /a/ | 0.39% | 0.37% | 0.23% |
| /ɾ/ | 0.35% | 0.67%* | 0.26% |
| /Q/ | 0.25% | 0.01% | 0.05% |
| /ɸ/ | 0.21% | 0.09%* | 0.11% |
| /t͡ɕ/ | 0.19% | 0.10% | 0.07% |
| /d͡ʒ/ | 0.19% | 0.11% | 0.14% |
| /d/ | 0.19% | 0.50% | 0.26% |
| /ɯ/ | | 0.51%* | 0.29% |
| /o/ | | 0.27% | 0.18% |
| /s/ | | 0.10% | 0.07% |

**Table 16. Confusion matrix for the MRF trained on all official Japanese Pokémon names and tested on all elicited samples.**

| | | Classification | |
|---|---|---|---|
| | | Pre-evolution | Post-evolution |
| Sample | Pre-evolution | 3055 | 1510 |
| | Post-evolution | 1283 | 2855 |

**Table 17. Confusion matrix for the MRF trained on all elicited samples and tested on all official Japanese Poké-mon names.**

| | | Classification | |
|---|---|---|---|
| | | Pre-evolution | Post-evolution |
| Sample | Pre-evolution | 1436 | 1291 |
| | Post-evolution | 585 | 2340 |

and elicited datasets on the Elicited and Japanese MRFs, respectively. Table 16 presents the RF trained on the official Japanese Pokémon names and tested on the elicited samples. Table 17 shows the confusion matrix for the RF trained using the elicited samples and tested on the official names.

Table 18. Results of testing the Japanese MRF on the elicited data from Experiment 1 and the Elicited MRF on the Japanese data. This includes both MRFs that do not contain length as a feature (-L OOBM) and those that do (+L OOBM) as well as their standard deviation (-L OOBSD, +L OOBSD).

| Train data | Test data | -L OOB$^M$ | -L OOB$^{SD}$ | +L OOB$^M$ | +L OOB$^{SD}$ |
|---|---|---|---|---|---|
| Japanese | Elicited | 37.31% | 1.26% | 32.09% | 0.75% |
| Elicited | Japanese | 35.86% | 2.01% | 33.19% | 2.03% |

To explore the issue of overfitting further, we reconstructed the -L Japanese MRF; this time, however, we skipped the tuning process and used the default hyperparameter settings in the Ranger package. This was done because we considered that the most likely reason for overfitting was low variability in decision trees due to the hyperparameter settings suggested by the tuning process. The untuned -L Japanese MRF (OOB error $M = 35.94\%$, $SD = 1.75\%$) was less accurate than the tuned MRF (OOB error $M = 34.07\%$, $SD = 2.4\%$), but the standard deviation was lower, suggesting that overfitting was less prevalent in individual RFs. We then recreated the untuned MRF using the entirety of the official Japanese names and tested it on the entirety of the elicited names and found the same pattern whereby the untuned MRF (OOB error $M = 38.24\%$, $SD = 0.42\%$) was less accurate, but more stable than the tuned MRF (OOB error $M = 37.31\%$, $SD = 1.26\%$) presented in Table 18.

We considered a potential alternative explanation for the high standard deviation in tuned MRFs; that variability caused by the randomization of subset splits may be explained by an over/under-representation of pre-/post-evolution Pokémon in the testing/training subsets. A simple regression model was constructed to predict the effect of increased post-evolution Pokémon in the testing subset on OOB error for all the Japanese, Chinese and Korean RFs taken from the MRFs. Elicitation data was not included because the distribution of samples to pre-/post-evolution categories is different in the elicited responses. No correlation between distribution in subsets and OOB error was observed, $F(1,25) = 0.31$, $p = 0.581$, $R^2 = 0.01$. We must therefore consider that the variability in accuracy when randomizing the subsets is most likely due to overfitting resulting from low variability in decision trees.

In the classification experiment, 119 Japanese participants each classified twenty names into either pre- or post-evolution categories. The twenty names were taken from 100 randomly selected samples from the results of Experiment 1. The participants were reasonably accurate ($M = 61.58\%$, $SD = 17.84\%$) at assigning the elicited Pokémon names to pre- and post-evolution categories. This assessment was based on the individual responses taken from their mean accuracy. This is arguably an unfair assessment of human ability, given that sound symbolic associations are decided upon by speech communities, not individual speakers. In RFs constructed for classification tasks, each decision tree votes for the classification of samples. The RF chooses the classification based on majority voting. To apply this method to the results of the classification experiment, we treated each response as a vote and examined the results of a majority vote analysis. Put simply, we examined the mode rather than the mean for each sample. Using majority voting, the participants in the classification experiment were able to accurately classify 71% of the samples. The same 100 samples were then tested using each RF in the MRF constructed with the official Japanese names. The MRF was able to accurately classify the samples far more accurately than the humans, correctly classifying samples 75.88% of the time (OOB error $M = 24.12\%$, $SD = 1.61\%$).

## Discussion

Experiment 2 was designed to test whether machine learning algorithms perform on par with humans, though it may not be immediately clear which of the MRFs presented in Table 18

should be used as a fair yardstick for the accuracy of the algorithms. We consider the results of the MRF trained using the length and sounds of all of the official Japanese Pokémon names and tested on the 100 samples used in Experiment 2 (OOB error $M$ = 24.12%) as the fairest measure for the performance of the algorithms, because they were trained on the maximum amount of information available that was also available to the human participants and tested on the same samples used in Experiment 2. Converting the responses of the human participants to OOB error shows us that the human participants (OOB error $M$ = 38.42%, $SD$ = 17.84%) were far less accurate than the algorithm (OOB error $M$ = 20.12%, $SD$ = 1.61%), even when using the majority vote method (OOB error = 29%).

The finding that the algorithms were more accurate than individual participants at classifying Pokémon is unintuitive, particularly given the limited data upon which the MRFs were trained. One interpretation of this finding is that human participants do not give their best effort all the time, while machine learning algorithms do. This lack of effort may come down to a lack of motivation, not taking the survey seriously, or any number of other factors that are simply impossible to take into account. However, we contend that this does not account for the entirety of the difference in classification accuracy for the following reasons. Firstly, the categorisation experiment was voluntary; participants were not rewarded monetarily or otherwise for their participation. While the printed handouts were distributed prior to classes, the students were not given any time in class to complete the experiment. It was done entirely in their own time. Additionally, the task was brief, taking around 2–3 minutes to complete. Lastly, the subject matter was specifically chosen because it was appealing and familiar to the population sample. Based on these factors, we expect that participant interest would have been high and that many participants would have been invested in the experiment.

Therefore, we believe that another interpretation may better explain the difference between participant and algorithm accuracy. That humans are susceptible to cognitive biases while machine learning algorithms are not. For example, humans will often apply oversimplified images or ideas to types of people or things, this is known as stereotyping. Through the lens of RFs, stereotyping is the overapplication of a feature to a category. Other cognitive biases suggest that humans do not intuitively understand probabilities, this is important given that sound symbolism is stochastic, not deterministic [24]. These biases include the *recency bias* (also known as the *availability bias*) which is the expectation that events that have occurred recently will reoccur regardless of their probability and the *conjunction fallacy* which is the assumption that a specific condition is more probable than a general one even when said specific condition includes the general condition [55]. Indeed, other studies have shown that machine learning algorithms can outperform humans (see [56] for a recent review). For example, McKinney et al. [57] presented a machine learning algorithm that outperformed six expert readers of mammographs in breast cancer prediction performance. Compared to the expert radiologists, the algorithm showed an absolute reduction in both false positives and false negatives. Given the nature of the task and the human participants, we can reasonably safely assume that the difference in performance was not based on disinterest or lack of motivation on the part of the radiologists. We must therefore consider that the OOB error difference between the human participants and the algorithm in this study is potentially due to a difference in learning efficiency and the application of that learning.

Length was omitted from the RFs presented in Experiment 1 because we wanted to isolate the feature importance of speech sounds, and the descriptive statistics suggested that Length was going to be an important feature that may mask the importance of weaker features. Indeed, Length was found to be important in all the MRFs. It was most important in the Elicited (6.99%) and Japanese (5.47%) MRFs which suggest that word length carries a considerable amount of sound symbolic information in Japanese. It was less important in the Korean

(2.44%) and Chinese (0.51%) MRFs. Isolating Length from the other features in Experiment 1 and introducing it into the RFs in Experiment 2 uncovered the issue of overfitting that led to the use of MRFs. The -L Japanese RF in Experiment 1 (OOB 29.05%) performed better than the +L Japanese RF in Experiment 2 (OOB 30.95%), despite Length being a highly important feature (4.54%) in the +L Japanese RF. The most likely explanation for overfitting is that there was little variability in decision trees. This hypothesis was tested by recreating the Japanese RFs using the default hyperparameter settings in the Ranger package. Running the untuned MRFs resulted in more stable RFs that were only slightly less accurate than their tuned counterparts. This finding supports our hypothesis that overfitting in Experiment 1 was the result of a lack of variability in decision trees. The lack of decision tree variability is likely due to a high number of features being examined at each node (mtry) which was suggested by the tuning process due to the large percentage of null values in the dataset (82.26%).

A potential solution to this issue was explored, which involved constructing each RF using the default hyperparameter settings; however, this resulted in an increased OOB error in all cases. Another potential solution would be to reduce the number of null values by reporting on phonological features rather than the sounds themselves. This would reduce both the number of null values and the number of features resulting in a less fine-grained data resolution. Instead, we constructed MRFs made up of independent RFs with different starting values for the randomisation of both the splitting of data into subsets and the RFs themselves. At first glance, MRFs may appear to be stacked RFs (SRFs: [58]), but this is not the case. Stacking [37] is a method of improving algorithm accuracy by combining weaker models into a super learner [59]. For example, Hänsch [58] sequentially adds RFs to SRFs using the estimates of earlier RFs to improve the accuracy of the final model. Our method is more like k-fold cross-validation which involves randomly dividing the data into k groups, or folds, and then recombining the data by way of a partial Latin square to create multiple training/testing subsets which are then used for constructing and testing multiple iterations of the algorithm [60]. K-fold cross-validation was not used in the present study because if the user adheres to the two-thirds subset rule, they are limited in choice for the number of iterations.

## Conclusion

The present study builds and tests machine learning algorithms using the names of Pokémon. Those algorithms are constructed to classify Pokémon into pre- and post-evolution categories. In Experiment 1, the algorithms are constructed using the speech sounds that make up Japanese, Chinese, and Korean Pokémon names. The feature importance calculations of these algorithms show that while some sound-symbolic patterns hold across languages, many important features are unique to each language. Experiment 1 also includes an elicitation experiment whereby Japanese participants named previously unseen Pokémon. We then construct RFs using the entirety of the official Japanese Pokémon name data and the elicited responses and test them on their opposite dataset. The OOB error of these tests shows that the sound symbolic patterns in these datasets are reasonably similar, suggesting that either those sound symbolic patterns already exist in the Japanese language, or the participants are familiar with Pokémon naming conventions. Previous studies have shown no correlation between Pokemon familiarity and sound symbolism effect size in nonce-word Pokémonastic experiments [61,62], suggesting that their results were not driven by existing knowledge of Pokémon names. In Experiment 2, all algorithms are reconstructed to include name length as a feature. This uncovers an issue of overfitting, which we resolve using MRFs. The performance of the MRFs is then measured against the performance of Japanese participants. The MRFs are shown to perform more accurately than humans.

RFs are said to be appropriate for "small N, high p" datasets [63], such as those found in the present study. However, Experiment 2 uncovers a clear case of overfitting in Experiment 1. The RFs constructed with length as a feature showed that length was important, yet this importance was not always reflected in OOB error. For example, the Japanese +L MRF (OOB = 31.69%) performed worse than the -L RF in Experiment 1 (OOB = 29.05%). Given that length was found to be important in the MRFs, this suggests that the individual RFs were overfitting because of the lack of variability in decision trees. Further evidence for this can be found in the difference between the accuracy of the RF trained on official Japanese Pokémon names to classify Elicited names (OOB = 38.57%) and the -L MRF trained on official Japanese Pokémon names to classify Elicited names ($OOB^M$ = 37.31%). In other words, the Japanese RF in Experiment 1 was more accurate than the Japanese MRF at classifying its own testing subset but less accurate at classifying the elicited samples because its function was too closely aligned to the initial dataset, resulting in a reduced capacity to classify external samples.

Sound symbolism is the study of systematic relationships between sounds and meanings. These relationships are not deterministic but rather stochastic, so they need to be observed through a statistical analysis. This paper details random forest algorithms that learn from these stochastic relationships and apply that learning to a classification task. Said task is the classification of Pokémon into pre- and post-evolution categories. This finding has important implications for the Natural Language Processing field of research, adding to the findings of Winter and Perlman [11] and showing that machine learning algorithms can make classification decisions driven (at least mostly) by sound symbolic principles, and should do so if the goal of an algorithm is to understand and use language the same way that humans do. The algorithms show how they make their classification decisions using feature importance, which is a useful metric for measuring the sound symbolic qualities of specific linguistic features. This is particularly useful when assessing universal sound-symbolic patterns. The present paper also exposes an issue of overfitting inherent in random forests constructed using decision trees with low variability. This issue is resolved by randomizing training and testing subset splits across multiple random forests. The machine learning algorithms are shown to be efficient learners in this task, achieving a higher classification accuracy than the human participants, despite having access to a limited number of samples from which to learn.

## Author Contributions

**Conceptualization:** Alexander James Kilpatrick, Shigeto Kawahara.

**Data curation:** Alexander James Kilpatrick.

**Formal analysis:** Alexander James Kilpatrick.

**Funding acquisition:** Alexander James Kilpatrick.

**Investigation:** Alexander James Kilpatrick.

**Methodology:** Alexander James Kilpatrick.

**Project administration:** Alexander James Kilpatrick.

**Resources:** Alexander James Kilpatrick.

**Software:** Alexander James Kilpatrick.

**Supervision:** Alexander James Kilpatrick.

**Validation:** Alexander James Kilpatrick.

**Visualization:** Alexander James Kilpatrick.

**Writing – original draft:** Alexander James Kilpatrick, Aleksandra Ćwiek.

**Writing – review & editing:** Alexander James Kilpatrick, Aleksandra Ćwiek, Shigeto Kawahara.

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
