## [Decision Letter · Decision Letter 0]

10 Oct 2022

PONE-D-22-19482Random Forests, Sound Symbolism and Pokémon EvolutionPLOS ONE

Dear Dr. Kilpatrick,

Thank you for submitting your manuscript to PLOS ONE. After careful consideration, we feel that it has merit but does not fully meet PLOS ONE’s publication criteria as it currently stands. Therefore, we invite you to submit a revised version of the manuscript that addresses the points raised during the review process.

We look forward to receiving your revised manuscript.

Kind regards,

Maki Sakamoto, Ph.D

Academic Editor

PLOS ONE

Journal Requirements:

a) Did participants provide their written or verbal informed consent to participate in this study?

3. We note you have included a table to which you do not refer in the text of your manuscript. Please ensure that you refer to Tables 3 and 4 in your text; if accepted, production will need this reference to link the reader to the Table.

4. Please ensure that you refer to Figures 4 and 5 in your text as, if accepted, production will need this reference to link the reader to the figure.

Reviewers' comments:

Reviewer's Responses to Questions

**Comments to the Author**

1. Is the manuscript technically sound, and do the data support the conclusions?

Reviewer #1: Yes

Reviewer #2: No

Reviewer #3: Partly

2. Has the statistical analysis been performed appropriately and rigorously? 

Reviewer #1: I Don't Know

Reviewer #2: No

Reviewer #3: I Don't Know

3. Have the authors made all data underlying the findings in their manuscript fully available?

Reviewer #1: Yes

Reviewer #2: No

Reviewer #3: No

4. Is the manuscript presented in an intelligible fashion and written in standard English?

Reviewer #1: Yes

Reviewer #2: No

Reviewer #3: Yes

5. Review Comments to the Author

Reviewer #1: 1. “Electabuzz” should be in italics. (p.6 in the Experiment 1 Section)

2. The original term “out-of-bag” for “OOB” should be introduced when it first appears. (p.9)

3. feature important => feature importance (Line 7, p.9)

4. six RFs conducted for Experiment 1 => six RFs constructed for Experiment 1 (the 2nd paragraph at p.9)

Reviewer #2: This study is to distinguish the names of characters of Pokemon as pre-evolved or post-evolved by means of machine learning technique (Random Forest), and to compare this performance of the machine learning with humans. The authors concluded that this machine learning method is efficient to learn “sound symbolism” and can classify samples with greater accuracy than the human participants.

I agree that the machine learning can categorize the pre- or post-evolved names somehow, but I disagree that this can learn the concept of “sound symbolism.” It seems to me that there are huge leaps between learning the pre- or post-evolved Pokemon names and learning the concept of sound symbolism. In this paper, the categorization performance of Pokemon’s pre- or post-evolved names by Japanese university students was set as humans’ standard performance for understanding sound symbolism. However, most Japanese students already have much knowledge about the Pokemon (knowledge contamination) so their categorization performances were not simply driven by the concept of sound symbolism as the author expected. The authors failed to clarify this issue.

The authors concluded in Experiment 2 like “the RFs showed the better accuracy than human participants” and discussed the reason of this issue. But the authors ascribed this issue to the experimental settings including participants’ attitudes. If they really consider so, they should conduct the modified experiment immediately.

Overall, this paper was quite redundant and difficult to read through. There was no information about the data acquired in their experiments, the training accuracy of RF, the effect size, and statistical analysis for the observed data.

Reviewer #3: We know that sound symbolism is not a strong property.

Therefore, we recognise that the experimental results presented in this paper suggest a statistical tendency.

However, before evaluating the experimental results, there are some unclear descriptions, which make the experimental results and the discussion unreliable.

The authors use the same approach as Winter and Perlman [11] for two-class classification.

When class classification is performed, a confusion matrix is usually presented.

Then, in addition to the accuracy, the authors show precision, recall, true positive rate, false negative rate or F-measure.

In particular, F-measure should be shown, the classification results for each class should be discussed based on accuracy, F-measure and so on.

The authors also introduce a feature importance, but the definition is not clear.

It is not clear what exactly "permutation" is.

A definition formula to calculate feature importance should be given.

Furthermore, the authors seem to have conducted a statistical test to show the p-values.

What is the statistical test employed in this paper?

6. PLOS authors have the option to publish the peer review history of their article (what does this mean?). If published, this will include your full peer review and any attached files.

Reviewer #1: No

Reviewer #2: No

Reviewer #3: No

---

## [Decision Letter · Decision Letter 1]

6 Dec 2022

Random Forests, Sound Symbolism and Pokémon Evolution

PONE-D-22-19482R1

Dear Dr. Kilpatrick,

We’re pleased to inform you that your manuscript has been judged scientifically suitable for publication and will be formally accepted for publication once it meets all outstanding technical requirements.

Kind regards,

Maki Sakamoto, Ph.D

Academic Editor

PLOS ONE

Additional Editor Comments (optional):

Reviewers' comments:

Reviewer's Responses to Questions

**Comments to the Author**

1. If the authors have adequately addressed your comments raised in a previous round of review and you feel that this manuscript is now acceptable for publication, you may indicate that here to bypass the “Comments to the Author” section, enter your conflict of interest statement in the “Confidential to Editor” section, and submit your "Accept" recommendation.

Reviewer #1: All comments have been addressed

Reviewer #3: All comments have been addressed

2. Is the manuscript technically sound, and do the data support the conclusions?

Reviewer #1: Yes

Reviewer #3: Yes

3. Has the statistical analysis been performed appropriately and rigorously? 

Reviewer #1: I Don't Know

Reviewer #3: Yes

4. Have the authors made all data underlying the findings in their manuscript fully available?

Reviewer #1: Yes

Reviewer #3: Yes

5. Is the manuscript presented in an intelligible fashion and written in standard English?

Reviewer #1: Yes

Reviewer #3: (No Response)

6. Review Comments to the Author

Reviewer #1: There is an error I have spotted at Line 598: previous => Previous (The initial 'p' should be a capital letter.)

Reviewer #3: Confusion matrixes are shown in the revised version. They make clear the classification results. If possible F-measures also should be shown. It would be better if the authors discussed the classification results based on the F-measure.

7. PLOS authors have the option to publish the peer review history of their article (what does this mean?). If published, this will include your full peer review and any attached files.

Reviewer #1: No

Reviewer #3: No

---

## [Editor Report · Acceptance letter]

9 Dec 2022

PONE-D-22-19482R1 

Random Forests, Sound Symbolism and Pokémon Evolution 

Dear Dr. Kilpatrick:

I'm pleased to inform you that your manuscript has been deemed suitable for publication in PLOS ONE. Congratulations! Your manuscript is now with our production department. 

Kind regards, 

on behalf of

Dr. Maki Sakamoto 

Academic Editor

PLOS ONE